# Molecular-level insights into the electronic effects in platinum-catalyzed carbon monoxide oxidation

Wenyao Chen[1], Junbo Cao[1], Jia Yang[2], Yueqiang Cao ![ORCID][1], Hao Zhang[3,4], Zheng Jiang ![ORCID][4,5], Jing Zhang[1], Gang Qian[1], Xinggui Zhou[1], De Chen[2✉], Weikang Yuan[1] & Xuezhi Duan[1✉]

A molecular-level understanding of how the electronic structure of metal center tunes the catalytic behaviors remains a grand challenge in heterogeneous catalysis. Herein, we report an unconventional kinetics strategy for bridging the microscopic metal electronic structure and the macroscopic steady-state rate for CO oxidation over Pt catalysts. X-ray absorption and photoelectron spectroscopy as well as electron paramagnetic resonance investigations unambiguously reveal the tunable Pt electronic structures with well-designed carbon support surface chemistry. Diminishing the electron density of Pt consolidates the CO-assisted $O_2$ dissociation pathway via the O*-O-C*-O intermediate directly observed by isotopic labeling studies and rationalized by density-functional theory calculations. A combined steady-state isotopic transient kinetic and in situ electronic analyses identifies Pt charge as the kinetics indicators by being closely related to the frequency factor, site coverage, and activation energy. Further incorporation of catalyst structural parameters yields a novel model for quantifying the electronic effects and predicting the catalytic performance. These could serve as a benchmark of catalyst design by a comprehensive kinetics study at the molecular level.

[1] State Key Laboratory of Chemical Engineering, East China University of Science and Technology, 130 Meilong Road, 200237 Shanghai, China. [2] Department of Chemical Engineering, Norwegian University of Science and Technology, 7491 Trondheim, Norway. [3] Institute of Functional Nano & Soft Materials Laboratory (FUNSOM), Jiangsu Key Laboratory for Carbon-Based Functional Materials & Devices, Soochow University, 215123 Suzhou, China. [4] Shanghai Institute of Applied Physics, Chinese Academy of Sciences, 201800 Shanghai, China. [5] Shanghai Synchrotron Radiation Facility, Zhangjiang Lab, Shanghai Advanced Research Institute, Chinese Academy of Sciences, 201210 Shanghai, China. ✉email: chen@nt.ntnu.no; xzduan@ecust.edu.cn

The electronic structure of transition metal catalysts harnesses the configuration and energy of the reaction species and transition states[1,2]. Taming the electronic structure of metal centers to coordinate their adsorption and activation toward optimized catalytic performance has emerged as an ideal scenario for energy-efficient catalysis[3,4]. Although various quantitative trends have been predicted using theoretical calculations[5,6], the experimental insights are still limited to qualitative descriptions because of the entangled structural variations encountered in realistic catalytic process[7,8], such as metal particle morphology/composition, support reducibility/polarity and acidity/basicity, which could yield inconclusive and even contradictory conclusions. Exemplified by Pt-catalyzed CO oxidation, the most investigated reaction in both practical application and fundamental research[9,10], a unified picture of Pt electronic effects remains debated: negatively charged Pt nanoparticles supported on $WO_3$[11], $TiO_2$[12,13], $MOF$[14], $Co-B/TiO_2$[15], and $CeO_2$[16] significantly promote CO oxidation, whereas an opposite trend was observed for $Al_2O_3$[17], $CNT$[18], and $SiO_2$[19]. These phenomena are reasonably arising from the transition from reducible to inert substrates as catalyst supports with various surface and interface properties, giving rise to different kinetics and reaction mechanisms[20–22]. Hence, the prerequisite to reveal the nature of the electronic effects relies on the exclusion of other factors to quantify how it tunes the reaction pathway and related kinetics.

Heterogeneous catalysis is a typical kinetics phenomenon[23], and chemical kinetics analysis has been extensively studied for the identification of critical factors for catalyst design[24,25]. Specifically, microkinetic modeling provides a powerful tool for disclosing critical reaction intermediates and rate-determining elementary reactions regarding the surface chemistry occurring on the surface of catalyst[26–28]. However, a rigorous understanding of metal catalyst functionality is still delayed by the strongly approximated description of metal sites, and the effects of their structural properties are usually neglected in state-of-the-art microkinetic modeling[29]. As such, the derived models are based on an abstract and static concept of the "catalyst material", normally denoted as a free site "*", thus giving rise to ambiguous relationships between the microscopic properties of metal sites and the macroscopic catalytic performance[29,30]. In this regard, obtaining in situ kinetics information on the adsorption and activation of reaction species, and further combining it with the isotopic studies, characterization results, and theoretical calculations could potentially allow for the bridging of this microscopic-to-macroscopic transition. Therefore, based on the above discussion, the disentanglement of metal electronic structures for identifying the kinetics indicator and reaction pathway, and incorporating them further to establish a simple yet quantitative principle or model represents two highly desirable goals in the development of microkinetic modeling to rationalize catalyst design at the molecular level.

In this study, we conducted a joint experimental and theoretical study to quantify the adsorption and activation of surface species on metal electrons over an unconventional kinetics strategy, aiming to bridge the upscaling gap between the microscopic fingerprints of active sites and the macroscopic catalytic performance: As shown in Fig. 1, we firstly designed and prepared a platform of well-defined Pt catalysts with controlled electronic structures, and verified it via microscopy and spectroscopy characterization. Thereafter, a combination of isotopic labeling techniques and density-functional theory (DFT) calculations was employed to identify the reaction pathway as well as kinetics indicator by correlating with the catalytic performance. Lastly, steady-state isotopic transient kinetic analysis (SSITKA) and in situ XPS measurements were conducted to obtain the kinetics behaviors under operando conditions for kinetics modeling. This in situ kinetics strategy for identifying reaction pathways and kinetics indicators to establish the new model could predict the catalytic performance and be extended to the designing of other metal catalysts.

## Results

### Designing and manipulating of Pt catalysts with tunable electronic properties.
In comparison with conventional metal oxides and zeolites, conductive carbon supports endow the catalysts with a unique and effective electron transfer system[31,32]. To obtain the similar structured Pt catalysts with tunable electronic structures, versatile carbon nanotubes (CNT) were employed as catalyst supports and they were activated via mixed acid oxidation, and were then annealed under an inert atmosphere at elevated temperatures (200, 400, 600, 800, and 1000 °C), as depicted in Fig. 2a. The $N_2$-physisorption results in Supplementary Table 1 indicate that the covalent functionalization and high-temperature annealing had slight influences on the morphology, surface area, and porosity of the CNT. In contrast, the types and concentrations of oxygen-containing groups (OCGs) over the CNT surface vary significantly with the heat treatment temperature owing to their different thermal stabilities, which could be employed as ligands to tailor the electronic properties of metal centers. Therefore, the as-obtained partially deoxygenated CNT (i.e., CNT-0, CNT-200, CNT-400, CNT-600, CNT-800, and CNT-1000) were characterized via Raman spectroscopy and thermogravimetric analysis (TGA) to investigate their structural evolution with temperature.

As shown in Supplementary Fig. 1, all these samples demonstrate five peaks fitting into the Raman spectra, in which the intensity ratio of D1 (1350 cm$^{-1}$) to G (1585 cm$^{-1}$) bands, $I_{D1}/I_G$, was employed to quantify the amount of surface defects as shown in Supplementary Fig. 2. Evidently, the $I_{D1}/I_G$ first decreases from 1.28 to 1.10 at 200 °C and then increases to 1.35 at 400 °C, which remains almost unchanged by increasing the temperature further. Moreover, the amounts of OCGs based on the TGA profiles (Supplementary Fig. 3) were 20.8, 12.7, 10.1, 8.0, 6.1, and 4.6 wt%, as shown in Supplementary Fig. 4. Based on the above analyses, the medium heat treatment (~400 °C) removes a majority of OCGs and simultaneously creates surface defects, whereas further increasing the temperature eliminates the remaining OCGs but has a negligible effect on the surface defects.

To acquire clear-cut information on the electronic properties of Pt, the CNT was impregnated with the same amount of Pt precursor (1.0 wt%). The high-resolution high-angle annular dark-field scanning transmission electron microscope (HAADF-STEM) images in Supplementary Fig. 5 reveal the homogeneous distribution of Pt nanoparticles, which is ascribed to the large external surface area of CNT and their strong interactions with OCGs and defects. The average Pt particle size ($d_{Pt}$) for these catalysts was determined to be 1.2–1.3 nm by averaging 200 random particles (Fig. 2b), and the highly dispersed Pt particles were confirmed by the $H_2$-chemisorption results in Supplementary Table 2. Moreover, the single crystallinity of the fcc structure of Pt nanoparticles is reflected in the atomic-scale images in Fig. 2c and Supplementary Figs. 6 and 7, and the corresponding fast Fourier transforms (FFT) and projections show the equilibrium shapes that could be truncated octahedrons, consistent with our previous study[33]. Hence, the almost identical Pt loadings, particle sizes, distributions, and morphologies indicate similar amounts of exposed Pt atoms for these catalysts, which will serve as the basis for the following mathematical modeling. Notably, not all the surface-exposed Pt atoms are active because CO could poison some of them with irreversible adsorption. Therefore, the number of Pt active sites was measured

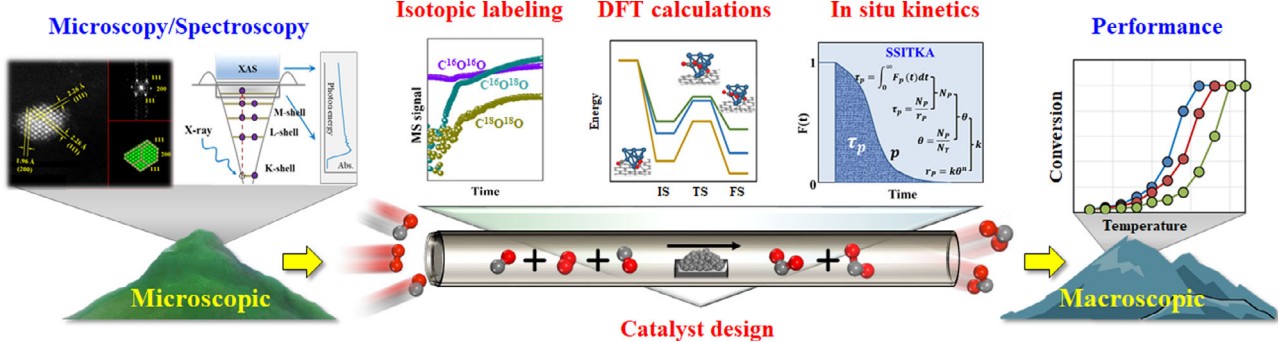

**Fig. 1 Strategy to bridge the microscopic-to-macroscopic transition.** Schematic diagram of bridging the upscaling gap between the microscopic fingerprints of active sites and the macroscopic catalytic performance based on (in situ) spectroscopy/microscopy characterization, isotopic labeling technique, DFT calculations and in situ kinetics measurements for CO oxidation.

**Fig. 2 Strategy for designing, preparing, and characterizing Pt catalysts. a** Schematic diagram of CNT activations via covalent functionalization and heat treatment. The yellow and light blue surfaces correspond to the electron increase and depletion zones, respectively. **b** The particle size and binding energy (B.E.) of Pt for Pt/CNT-0, Pt/CNT-200, Pt/CNT-400, Pt/CNT-600, Pt/CNT-800 and Pt/CNT-1000. **c** Aberration-corrected high-angle annular dark-field scanning transmission electron microscopy (HAADF-STEM) images, the corresponding fast Fourier transform (FFT) pattern, and projection of a truncated octahedron model of the Pt/CNT-600 catalyst. The EPR spectra (**d**), Pt L$_3$-edge XANES profiles (**e**), and the relationship between n$_{EWG}$/n$_{EDG}$ and Pt B.E., Pt charge as well as EPR intensity (**f**) for Pt/CNT-0, Pt/CNT-200, Pt/CNT-400, Pt/CNT-600, Pt/CNT-800, and Pt/CNT-1000.

using $^{12}C^{16}O$-$^{13}C^{16}O$ isotopic switches (Supplementary Fig. 8) according to the method described in our previous study[34], thus yielding the estimated Pt particle size further (Supplementary Table 2). Comparatively, these Pt catalysts exhibit slightly smaller or equivalent Pt particle sizes compared with those determined by HAADF-STEM, except for Pt/CNT-0. This could be because of the presence of a few irreversible CO adsorptions. As a result, the similar Pt particle sizes determined via either HAADF-STEM or $^{12}CO$-$^{13}CO$ isotopic switches indicate similar numbers for the Pt active sites over these catalysts, making them the ideal model catalysts for exclusively studying the effects of Pt electronic structures.

The electronic structures of the catalysts were first studied using electron paramagnetic resonance (EPR) and X-ray absorption near-edge structure (XANES) spectroscopy. As shown in Fig. 2d, the strong EPR signal with symmetric resonance for Pt/CNT-0 indicates abundant unpaired electrons on the catalyst surfaces. It is evident that the heat treatment significantly reduced the EPR signal intensity to a minimal at 600 °C, indicating the scarcity of electrons on the surface of Pt/CNT-600. Subsequently, the EPR signal becomes stronger again. Moreover, a shift in the $g$-value was observed to follow the same tendency as the signal intensity. Figure 2e shows the normalized Pt $L_{III}$-edge XANES spectra of these Pt catalysts, reference material $Pt^0$ (Pt foil) and $PtO_2$, in which an obvious peak at 11567 eV, known as the white-line (WL), represents electronic transition from $2p_{3/2}$ to $5d_{5/2}$ and $5d_{3/2}$ at the Pt $L_{III}$-edge[35]. Clearly, the raise of the white-line intensity by increasing heat treatment temperature indicates that the unoccupied $d$ states of Pt increased with a loss of its $5d$ electrons. Generally, this electron relocation could lead to a charge transfer between Pt and OCGs with a concomitant change in Pt valence state. Hence, the average surface charge of Pt in terms of its valence state could be calculated as +0.91, +0.93, +1.03, +1.05, +0.99, and +0.97 based on the relationship between the ionic valence and white-line intensity of the reference compounds (Pt foil and $PtO_2$)[3].

Generally, the changes in valence state by the bonding or hybridization between neighboring atoms give rise to core-level shift[36], which was further investigated by XPS. As shown in Supplementary Fig. 9, the XPS Pt $4f$ spectra were characterized by a typical doublet of Pt $4f_{7/2}$ and Pt $4f_{5/2}$, and deconvoluted into three pairs of doublets ($Pt^0$, $Pt^{2+}$, and $Pt^{4+}$) with similar percentages as summarized in Supplementary Table 3. In comparison, a significant shift in Pt binding energy (B.E.) was observed. For the $Pt^0$ $4f_{7/2}$ spin-orbit peak, the corresponding Pt B.E. continuously increased from 71.60 eV (Pt/CNT-0) to 71.90 eV (Pt/CNT-600), and then declined to 71.74 eV (Pt/CNT-1000), as depicted in Fig. 2b. Considering the high electron conductivity of CNT (Supplementary Table 4) in terms of the high percentage of $sp^2$-hybridized carbon with respect to $sp^3$-hybridized carbon (Supplementary Fig. 10) to neutralize the initial ion charge, the observed binding energy shift is mainly attributed to the electron transfer between Pt and CNT. As the Pt $4f$ level is not too deep and the wave-function overlap between the Pt $4f$ and $5d$ levels is significant, the variation in the Pt $5d$ states occupation number yields more pronounced effects in the determination of binding energy shift than that of total number of electrons[35]. As a result, both XANES and XPS analyses confirm the significant loss of $5d$ state electrons of Pt over CNT-600 upon bonding/hybridization with OCGs from the inductive/resonance effects[37], which is vice versa for Pt over CNT-0.

To elucidate the underlying factors of electron transfer between Pt and CNT, the O 1 s spectra shown in Supplementary Fig. 11 were deconvoluted, which could be further categorized into electron-withdrawing groups (EWG), including carbonyl

(531.1 eV), ester (533.4 eV) and carboxyl (534.4 eV), as well as electron-donating groups (EDG) such as hydroxyl (532.1 eV) (Supplementary Table 5)[37,38]. Unexpectedly, the Pt B.E., Pt charge and EPR intensity exhibit almost linear dependences on the molar ratio of EWG to EDG ($n_{EWG}/n_{EDG}$) as depicted in Fig. 2f. Specifically, the surface-enriched EWG with respect to EDG over the CNT-600 captures more electrons from Pt particles to lower its electron density with the highest B.E., which is in contrast for Pt/CNT-0 with the lowest B.E. Hence, the high consistency among the Pt B.E., Pt charge, and EPR intensity in Fig. 2f clearly reveal the tunable Pt electronic structures, which could be fine-tuned by $n_{EWG}/n_{EDG}$ via controllable heat treatment.

**Reaction pathway modulation and identification of kinetics indicators.** Accordingly, various Pt model catalysts were prepared with a unique opportunity to individually assess the Pt electronic effects by excluding other factors. Herein, CO oxidation was selected as the prototypical reaction, where the effects of external and internal diffusion limitations were excluded (Supplementary Information). Figure 3a shows the CO conversion as a function of the reaction temperature, represented by a typical light-off process. As the $^{12}C^{16}O$-$^{13}C^{16}O$ isotopic switches were conducted at 100 °C, the reaction rates for these catalysts at the same temperature ($r_{100}$) were calculated and were compared in Fig. 3b, which further yielded the turnover frequency (TOF) based on HAADF-STEM and $^{12}CO$-$^{13}CO$ isotopic switches, that was $TOF_{Pt}$ and $TOF_{CO}$ as summarized in Supplementary Table 2. The further correlation in Fig. 3c indicates an almost one-to-one relationship between the catalytic activity and Pt charge.

To understand the Pt electronic effects, an isotopic labeling technique was employed to investigate the reaction pathway, which remains debated between the CO-assisted (Supplementary Fig. 12) and direct $O_2$ dissociation (Supplementary Fig. 13)[20,39,40]. As shown in Fig. 4a, and Supplementary Figs. 14, 15, and 16, the formation of $^{16}O^{18}O$ isotopologues from equimolecular $^{16}O_2$/$^{18}O_2$ mixtures for these catalysts followed the same trend as that without the catalyst (Supplementary Fig. 17) and the composition of the feed gas (Supplementary Fig. 18). As the quasi-equilibrated $O_2$ dissociation forms binomial isotopologue distributions (50% $^{16}O^{18}O$)[41], the absence of that suggests the sluggish dissociation/association of oxygen. Conversely, the detection of $C^{18}O^{18}O$ for these catalysts, specifically Pt/CNT-600, indicates an isotopic exchange within the reaction intermediate (such as $^{18}O^*$-$^{18}O$-$C^*$-$^{16}O$ to $^{16}O^*$-$^{18}O$-$C^*$-$^{18}O$), which would not occur for the atomic oxygen reacting with CO ($O^* + CO^* \rightarrow CO_2^* + {}^*$) as a result of direct $O_2$ dissociation. Moreover, the effects of CO conversion and reaction temperature on the reaction pathway were found to be significantly smaller compared to those of Pt electronic effects, based on the detection of $C^{18}O^{18}O$ in Supplementary Figs. 19 and 20. The Pt/CNT-600 catalyst exhibited a slight increase in the $^{16}O^{16}O$ signal during the switch to $C^{16}O + {}^{18}O_2$, indicating the presence of $^{16}O^*$ after the decomposition of $^{16}O^*$-$^{18}O$-$C^*$-$^{18}O$ and their subsequent association to yield $^{16}O_2$. Based on the above analyses, it can be deduced that the reaction pathway shifts to CO-assisted $O_2$ dissociation via the formation of OOCO by diminishing the electron density of Pt.

With the above compelling mechanisms, DFT calculations were carried out to determine the energy barriers and adsorption energies of the reactant species. Based on the XPS O 1 s spectra (Supplementary Fig. 11), the structural models of the above four OCGs-incorporated substrate-supported $Pt_{10}$ clusters (Pt-hydroxyl, Pt-carboxyl, Pt-carbonyl, and Pt-ester) were constructed for a comparative study with pure substrate-supported one (Pt-basal)

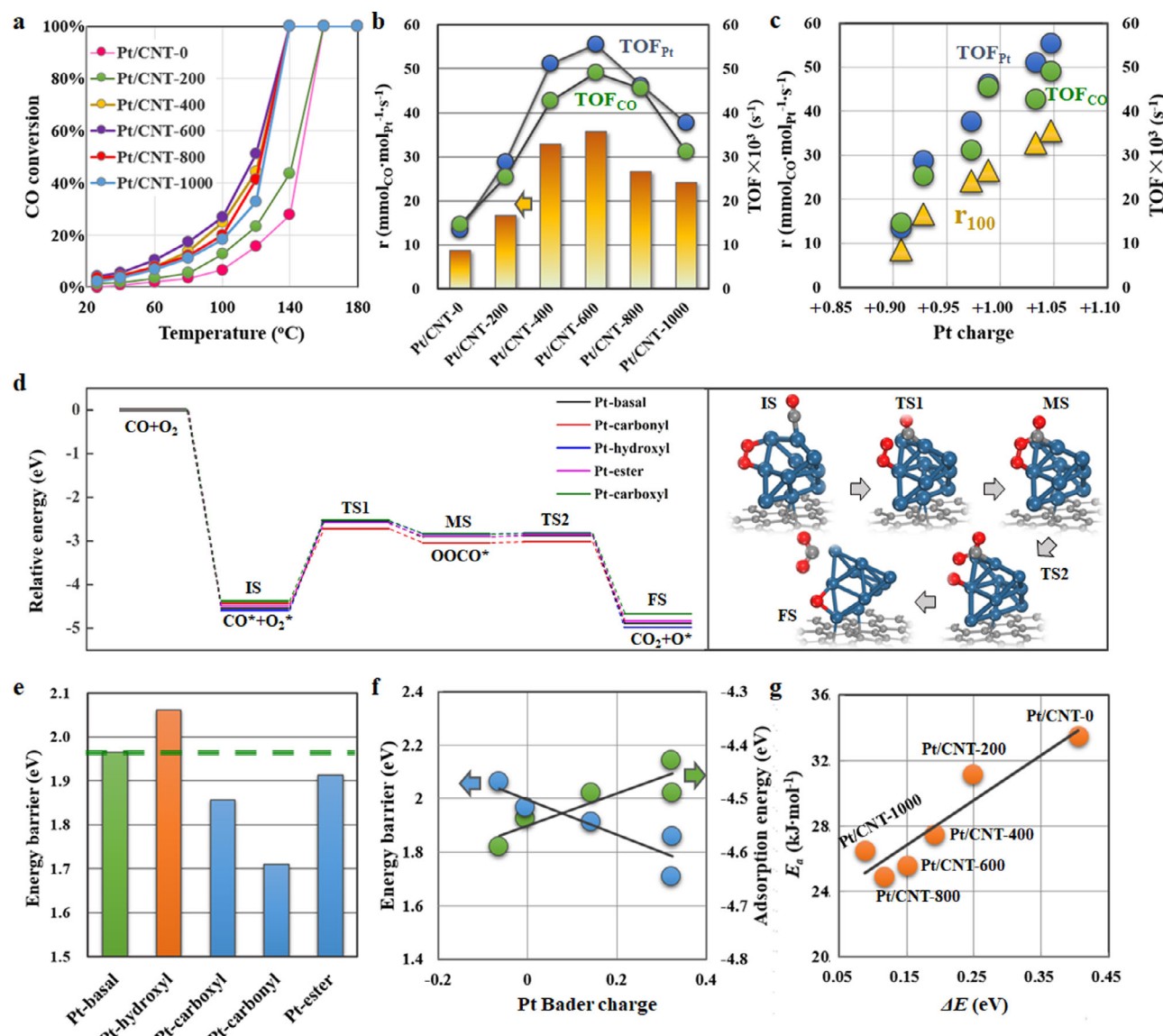

**Fig. 3 Catalytic testing and DFT calculations of CO oxidation. a** CO conversion as a function of temperature. The reaction rate ($r_{100}$) and the corresponding $TOF_{Pt}$ and $TOF_{CO}$ at 100 °C (**b**), as well as the relationships between $r_{100}$, $TOF_{Pt}$, $TOF_{CO}$ and Pt charge (**c**), for Pt/CNT-0, Pt/CNT-200, Pt/CNT-400, Pt/CNT-600, Pt/CNT-800 and Pt/CNT-1000. Potential energy diagram from DFT calculations and the schematic diagram of the related configurations over Pt-basal (**d**), and the corresponding energy barrier (**e**) for CO oxidation over Pt-basal, Pt-hydroxyl, Pt-carboxyl, Pt-carbonyl, and Pt-ester. **f** The relationships between energy barrier ($\Delta E_i$), adsorption energy ($E_{ads}$) and Pt Bader charge. **g** The relationship between the activation energy ($E_a$) and the proposed energy barrier ($\Delta E$), incorporating the influences of the above OCGs (Supplementary Information), for Pt/CNT-0, Pt/CNT-200, Pt/CNT-400, Pt/CNT-600, Pt/CNT-800, and Pt/CNT-1000.

as shown in Supplementary Fig. 21. As a result, the Bader charge analyses indicated a negligible electron transfer between Pt and pure substrate, whereas those of –0.06, 0.33, 0.32, and 0.14 e for Pt-hydroxyl, Pt-carboxyl, Pt-carbonyl, and Pt-ester, respectively. Accordingly, Fig. 3d shows the lowest energy pathways for $CO_2$ formation, and the corresponding optimized configurations are also displayed in Supplementary Figs. 22–25.

Evidently, the rate-determining step involves the formation of a peroxide-like complex (OOCO) from adsorbed CO and $O_2$, in which the energy barrier ($\Delta E_i$) follows the order of Pt-hydroxyl (2.06 eV) > Pt-basal (1.97 eV) > Pt-ester (1.91 eV) > Pt-carboxyl (1.86 eV) > Pt-carbonyl (1.71 eV), as shown in Fig. 3e, and this was qualitatively compared to the study by Ramasubramaniam et al.[42] Further correlation with the Pt Bader charge in Fig. 3f indicates an almost linear decline in the energy barrier ($\Delta E_i$) with the Pt Bader charge. Considering that the adsorption of reactants,

specifically CO, has been suggested as the key factor for this reaction, the effects of Pt charge on their adsorption were further compared in Fig. 3f. Obviously, the adsorption energy ($E_{ads}$) demonstrates an opposite trend that linearly increases with Pt Bader charge. This is consistent with a previous study showing that the strong adsorption of CO severely poisons Pt active sites, and a general consensus to improve the activity relies on weakening the Pt-CO bonding for CO desorption to provide more active sites for $O_2$ activation[43].

Notably, the above result is still speculative because of the lack of direct evidence on the activation and adsorption of reaction species under operando conditions. Moreover, it has been suggested in previous study that the changes of the electronic properties of metal may result in an increase of electron back-donation, but not necessarily a strengthening of the M-CO bond[12]. Therefore, in situ kinetics information to bridge the

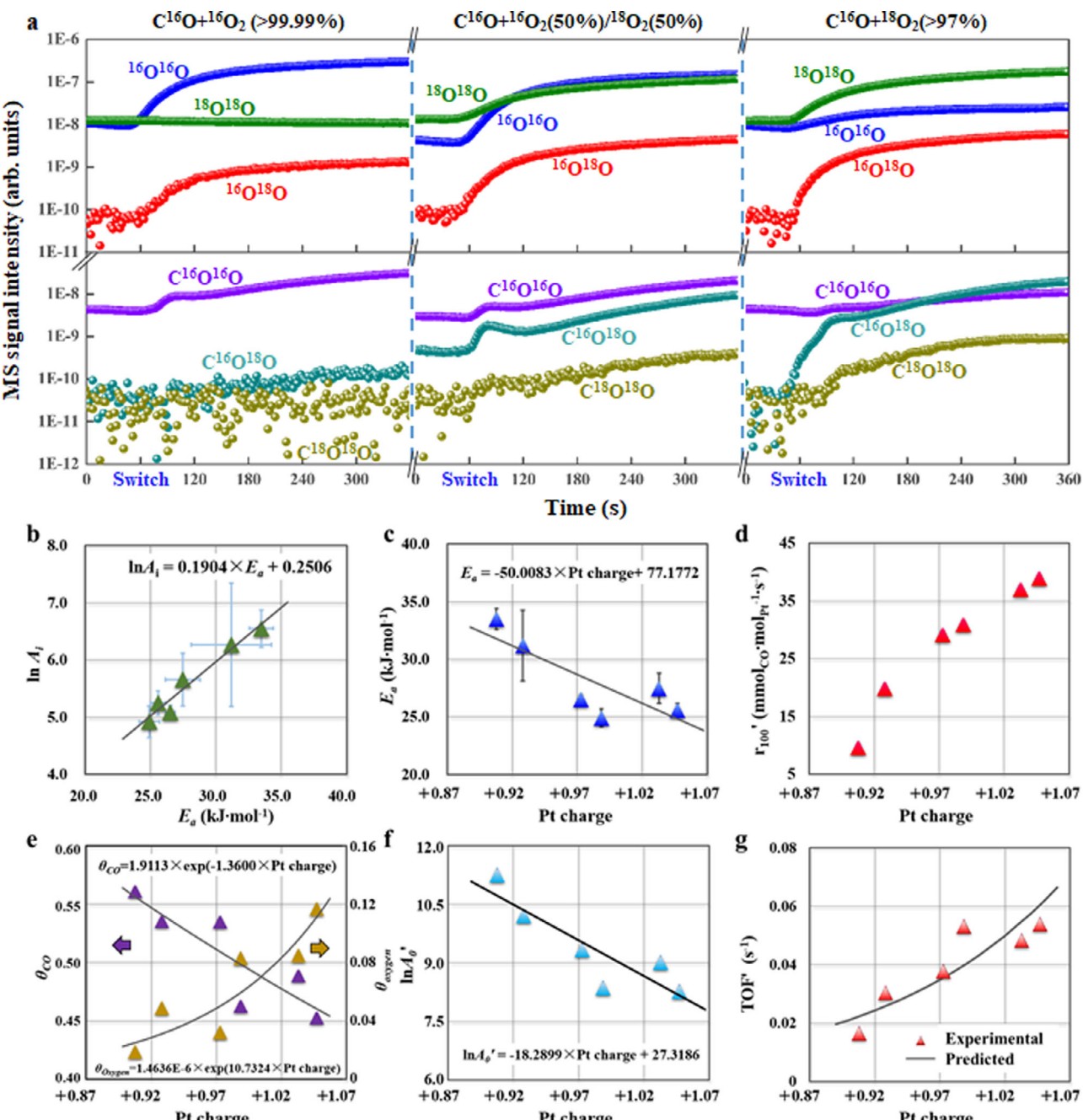

**Fig. 4 Isotopic labeling studies and SSITKA of CO oxidation. a** Mass spectrometry (MS) data collected for the Pt/CNT-600 catalyst during the switch from Ar to Ar+C$^{16}$O + $^{16}$O$_2$ (the purity is >99.99%), Ar+C$^{16}$O + $^{16}$O$_2$ (50%)/$^{18}$O$_2$ (50%), and Ar+C$^{16}$O + $^{18}$O$_2$ (the concentration of $^{18}$O$_2$ is > 97%) at an ambient pressure. **b** The relationship between logarithm of frequency factor (ln$A_i$) and activation energy ($E_a$). The $E_a$ (**c**), r$_{100}$' (**d**), site coverages of CO ($\theta_{CO}$) and oxygen ($\theta_{oxygen}$) (**e**), logarithm of frequency factor (ln$A_0$') (**f**), as well as the experimental and predicted TOF' (**g**) as a function of Pt charge for Pt/CNT-0, Pt/CNT-200, Pt/CNT-400, Pt/CNT-600, Pt/CNT-800, and Pt/CNT-1000. Reaction conditions: 100 °C, P$_{CO}$:P$_{O2}$:P$_{Ar}$ = 1:20:79, and 60,000 mL·g$_{cat}$$^{-1}$·h$^{-1}$. Error bars in **b** and **c** were calculated from the standard error of each linear fit presented in Supplementary Fig. 26.

microscopic electronic properties of the Pt active sites with the macroscopic catalytic performance of CO oxidation were obtained as follows. First, as shown in Fig. 3a, the CO conversion follows a typical light-off curve and the reaction rate in the intermediate region can be reasonably extracted for Arrhenius plots in Supplementary Fig. 26, yielding the activation energy ($E_a$) and the logarithm of frequency factor (ln$A_i$) listed in Supplementary Table 6. Unexpectedly, ln$A_i$ exhibits an almost linear dependence on $E_a$ as shown in Fig. 4b, indicating a significant kinetic compensation effect. According to the Cremer-Constable

relation, this relation could be interpreted as a thermodynamic balance between the activation entropy and activation enthalpy[44]. Based on the transition-state theory, these two kinetics parameters represent the adsorption and activation of reaction species, respectively. Therefore, the activation properties of the reactants ($E_a$) could be well correlated with the Pt electronic properties (Pt charge), as shown in Fig. 4c. To confirm this trend, the $E_a$ derived from the kinetics experiments was plotted with the proposed energy barrier ($\Delta E$) (Supplementary Information) in Fig. 3g. Interestingly, the almost linear relationship between $E_a$

and $\Delta E$, as well as the linear dependence of $\Delta E_i$ on the Pt Bader charge (Fig. 3f) further validates the above analyses on the linear correlation between $E_a$ and Pt charge. Hence, based on the experimental and theoretical study, the Pt charge and Pt Bader charge could serve as the activation indicators of $E_a$ and $\Delta E$, respectively.

On the other hand, to experimentally elucidate the influences of Pt electronic properties on reactants adsorption properties, steady-state isotopic transient kinetic analysis (SSITKA) was conducted to determine the site coverage of surface species and the intrinsic reaction rate under operando conditions. Typically, the reaction temperature and pressure were raised to 100 °C and 1.85 bar, respectively, which were maintained for another 2 h prior to measuring the reaction rate ($r_{100}'$) and TOF′, as shown in Fig. 4d and Supplementary Table 7. Specifically, the $^{12}CO$ response curves obtained by switching from $Ar/^{12}CO/O_2$ to $Kr/^{13}CO/O_2$ were recorded in Supplementary Fig. 27, based on which the amount of adsorbed CO ($N_{CO}$) and site coverage ($\theta_{CO}$) could be calculated, as summarized in Supplementary Table 7. Moreover, the $O_2$ site coverage ($\theta_{O_2}$) was investigated by switching from $Ar/CO/^{16}O_2$ to $Kr/CO/^{18}O_2$, as shown in Supplementary Fig. 28, which exhibits the almost identical $^{16}O_2$ and Ar response curves. This could be because of the limited $O_2$ adsorption on the catalyst surface. However, based on the formation of OOCO from adsorbed CO and $O_2$ to produce $CO_2$ and the direct reaction between the adsorbed O and CO to produce $CO_2$, the amount of adsorbed oxygen species (mainly $O_2$) could be estimated from the amount of generated $CO_2$ owing to the excess of CO over the Pt surface. As a result, the oxygen site coverage ($\theta_{oxygen}$) based on the $C^{16}O_2$ response curves by switching from $Ar/CO/^{16}O_2$ to $Kr/CO/^{18}O_2$ (Supplementary Fig. 29) were calculated, as summarized in Supplementary Table 7.

Similarly, inspired by the linear dependence of adsorption energy ($E_{ads}$) on Pt Bader charge (Fig. 3f), the adsorption behaviors of the reaction species ($\theta_{CO}$ and $\theta_{oxygen}$) were further correlated with the Pt charge by exponential functions (Fig. 4e). Evidently, $\theta_{CO}$ decreases with Pt charge, which coincides with the decreased adsorption energy by the Pt Bader charge, whereas $\theta_{oxygen}$ exhibits an opposite trend. Moreover, the reaction orders of CO and $O_2$ for the most positively charged Pt/CNT-600 and negatively charged Pt/CNT-0 were measured as shown in Supplementary Fig. 30. It is obvious that Pt/CNT-0 exhibits much lower CO reaction order of -0.58 compared with that of −0.10 for Pt/CNT-600, consistent with its higher CO site coverage. On the other hand, the almost same $O_2$ reaction orders around 0.9 further evidence the much lower $O_2$ site coverages for these catalysts. Hence, this above kinetics information provides a quantitative description of CO and oxygen adsorption with Pt charge, which coincides with the positively charged Pt weakening the adsorption of CO and providing more active sites for oxygen adsorption, as indicated by DFT calculations. Notably, despite of the above good relationships, the Pt electronic structure under real reaction conditions could be different from that under characterization, and its possible change should be considered. Therefore, the Pt/CNT-600 catalyst with the highest catalytic activity was chosen for in situ XPS measurements over a homemade equipment, which was treated in a reaction chamber and then transferred through a load-lock gate to the analysis chamber. Figure 5 depicts the corresponding XPS Pt 4f spectra in the reaction atmosphere at elevated temperatures. It is evident that either the B.E. or percentage of Pt species was almost identical up to a temperature of 150 °C, whereas further increasing the temperature resulted in partial oxidation of Pt, as observed by Bernasek and Naitabdi et al[45,46]. Based on the above discussion, the variation in Pt electronic structure during the kinetics measurement is minimal, and thus Pt charge could be

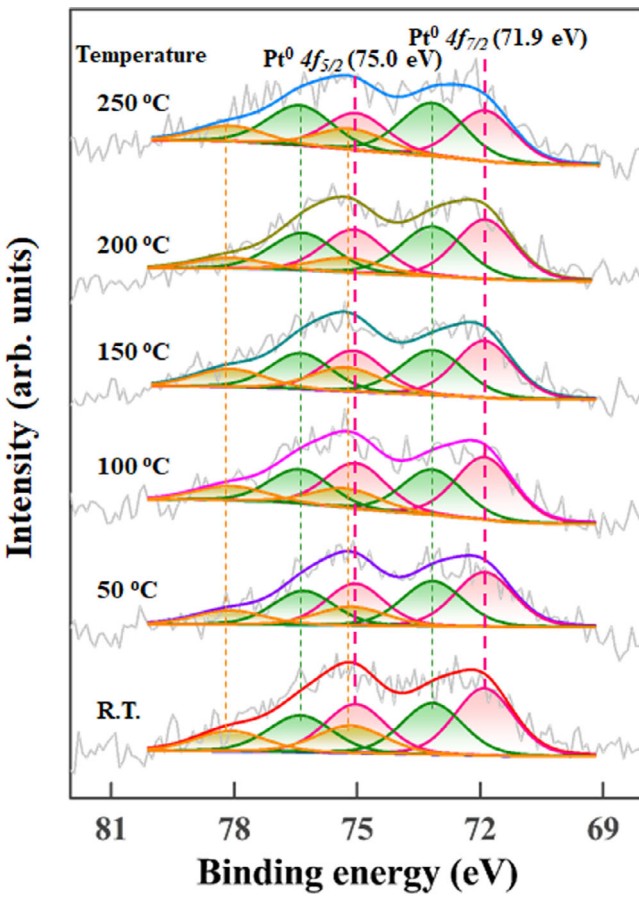

**Fig. 5 In situ electronic structure characterization.** In situ XPS Pt 4f spectra of Pt/CNT-600 under the reaction atmosphere at elevated temperature.

exclusively identified as an indicator of the adsorption and activation of reaction species over the catalyst surfaces under operando conditions.

**Molecular-scale kinetics modeling.** According to the above DFT calculations, the rate-determining step for CO oxidation for these Pt catalysts involves the formation of OOCO species from adsorbed CO and $O_2$ (Fig. 3d). As the reaction order for each reactant in an elementary step is equal to its stoichiometric coefficient, the reaction orders of CO and $O_2$ in this rate-determining step ($CO^* + O_2^* \rightarrow OOCO^* + *$) are determined to be 1. Therefore, the catalytic activity of the active site (TOF′) could be estimated using the expression of $TOF' = k' \times \theta_{CO} \times \theta_{oxygen}$, where $k'$ is the reaction rate constant. Combining the Arrhenius equation, $k' = A_0' \times \exp(-E_a/RT)$, further yields Eq. 1:

$$\ln A_0' = \ln TOF' - \ln \theta_{CO} - \ln \theta_{oxygen} + E_a/RT \quad (1)$$

As a result, $\ln A_0'$ could be calculated as summarized in Supplementary Table 7, and then further correlated with the Pt charge shown in Fig. 4f. According to the transition-state theory, $\ln A_0'$ is proportional to the activation entropy ($\Delta S^{0*}$), which mainly quantifies the freedom loss of reactant species in terms of binding strength with the catalyst. As the Pt charge could provide a good description of the activation (Fig. 4c) and adsorption behaviors (Fig. 4e, f) of the reactants, we attempted to combine the effects of the Pt charge on the activation and adsorption to derive a novel kinetics model to bridge the microscopic properties of Pt active sites and the macroscopic catalytic performance.

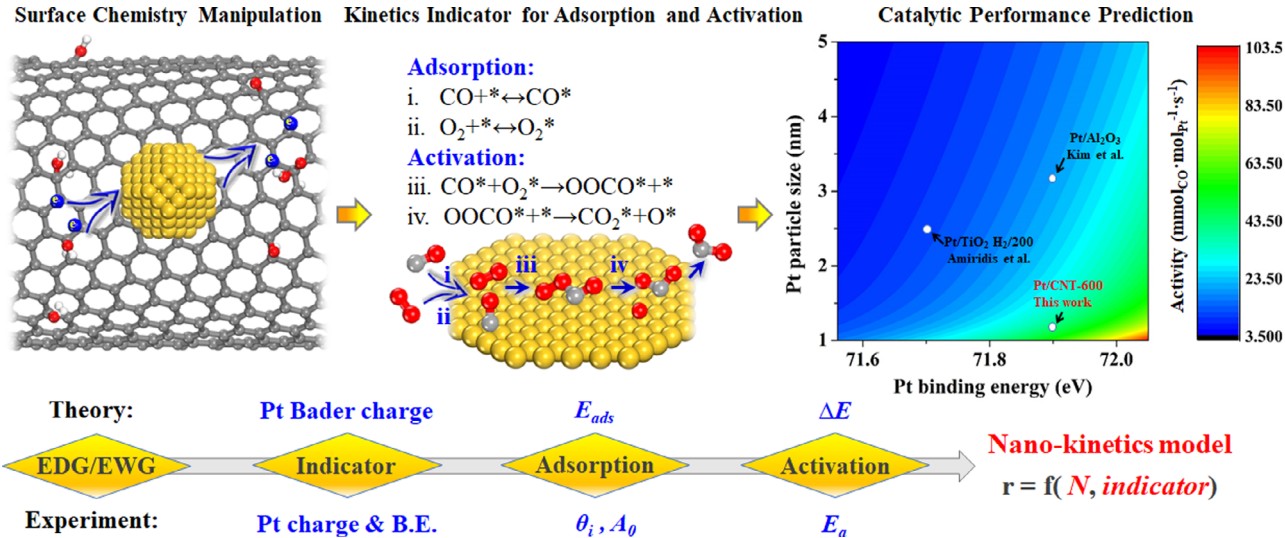

**Fig. 6 Model derivation of CO oxidation.** Schematic diagram of model derivation for Pt-catalyzed CO oxidation based on theoretical calculations and experimental investigations.

Combining the above linear relationships gives Eq. 2:

$$TOF\prime = a \times \exp(b \times Pt\ charge + c) \qquad (2)$$

where a, b, and c are determined in Supplementary Table 8 under the given conditions. As a result, the predicted TOF′ (solid line) based on Eq. 2 coincides with the experimental TOF′ (red triangle) in Fig. 4g. Similarly, as shown in Supplementary Fig. 31, employing Pt B.E. could also help derive the expression of TOF′ in good agreement with the experimental TOF′, thus validating the electronic structure of Pt as a kinetics indicator in the above analyses further. Considering the active site, the resultant catalytic performance is determined by both the quantity (N) and quality (TOF) of the active site, that is, $r = N \times TOF$ (Supplementary Information), which quantitatively correlates the Pt charge with the quality (TOF) under a similar quantity (N) of the active site. Furthermore, to incorporate the effects of N, the number of Pt active sites represented by Pt particle size was further calculated, as summarized in Supplementary Table 9. Hence, assuming negligible changes in TOF with N, the combined effects of N and TOF of Pt active sites could help derive a kinetics model, as summarized in Supplementary Table 10 and further schematized in Fig. 6. Evidently, this model links the electronic and structural properties of Pt active sites at the nanoscale with the macroscopic catalytic performance, demonstrating Pt B.E. or Pt charge as an experimentally accessible descriptor for predicting the catalytic performance, which could be used to search for the optimal Pt catalyst. The smaller Pt particles with high electron-deficiency supported on the EWG-rich CNT-600 exhibit much higher catalytic activity compared to those on other substrates at the similar reaction conditions (temperature and partial pressures), as shown in Fig. 6[12,47].

## Discussion

CO oxidation has been widely studied as a prototypical reaction to elucidate the reaction kinetics for the construction of the electronic structure-surface intermediates-catalytic performance relationship. Conventionally, kinetics studies on CO oxidation were dominated by macro- and micro-kinetics models based on the power law and the reaction mechanism as well as the elementary steps, respectively. With the advancement of computational chemistry and characterization techniques, the surface structure of metal site plays vital roles in catalysis science since

the discovery of structure-sensitive reactions, whose effects on reactivity are usually neglected in state-of-the-art microkinetic modeling[30]. Therefore, based on (in situ) spectroscopy/microscopy, isotopic labeling, DFT calculations, and in situ kinetics analysis, we presented an unconventional kinetics strategy for bridging this "materials gap" in traditionally kinetics analysis to correlate the microscopic electronic properties of active sites with the macroscopic catalytic performance as schematically depicted in Fig. 6. A platform of Pt catalysts with similar structural properties was provided, whose electronic properties were fine-tuned by manipulating the carbon surface chemistry ($n_{EDG}$/$n_{EWG}$). Inspired by DFT calculations on the linear relationship between the Pt Bader charge and adsorption energy ($E_{ads}$) as well as the activation barrier ($\Delta E$), the Pt charge was linearly correlated with the adsorption ($A_0$ and $\theta_i$) and activation behaviors ($E_a$) of the reaction species. Therefore, decoupling from the inherent structural and chemical complexity of carbon, the Pt charge is exclusively identified as the kinetics indicator of the active sites being the basis of nano-kinetics modeling, which can individually quantify the contributions of the microscopic electronic and structural properties of the active sites and predict the macroscopic catalytic performance. Potentially, such strategy focused on nanocatalysis could be extended to the design and manipulation of sub-nanocatalysis for sustainable chemical processes.

In summary, we report an unconventional kinetics strategy for quantifying the electronic effects at the molecular level for Pt-catalyzed CO oxidation, thus bridging the upscaling gap between the microscopic fingerprints of active sites and the macroscopic catalytic performance. Various Pt model catalysts with controllable electronic properties and uniform structural properties were prepared via covalent functionalization followed by heat activation of the carbon support. The reaction pathway was found to shift to CO-assisted $O_2$ dissociation via the formation of O*-O-C*-O by diminishing the electron density of the Pt particles. The quantitative relationships between the surface chemistry of the carbon support, electronic and structural properties of the Pt active sites, as well as in situ kinetics for CO oxidation enabled the identification of Pt charge as the kinetics indicator, linearly correlated with the adsorption (site coverage and frequency factor) and activation (activation energy) of the reaction species and rationalized by DFT calculations. Further incorporating kinetics

indicators with the number of Pt active sites initiates a novel model that offers a principle for designing and optimizing the Pt catalysts as well as predicting catalytic behaviors.

## Methods

**Catalyst preparation.** Pristine multi-walled CNT with a diameter of 50–90 nm and carbon basis of more than 95% were purchased from Sigma-Aldrich. To activate the CNT, they were treated with a mixture of 8 M H$_2$SO$_4$ (98%, Merck) and 8 M HNO$_3$ (65%, Merck) at 60 °C in an ultrasonic bath for 2 h. Thereafter, the oxidized CNT were filtered, washed with a large quantity of deionized water, and then dried overnight at 80 °C. The as-obtained CNT were divided into separate batches and treated at different temperatures, including 200, 400, 600, 800, and 1000 °C, in flowing Ar for 2 h. The thermally defunctionalized CNT is denoted as CNT-X, where X represents the temperature of the heat treatment. CNT-X was employed as support to prepare Pt catalysts by incipient wetness impregnation method. Typically, calculated amounts of H$_2$PtCl$_6$ (≥37.5% metal-based, Sigma-Aldrich) were dissolved in deionized water to obtain 1.0 wt% Pt loading. Thereafter, the catalyst precursors were aged at room temperature overnight and dried at 80 °C for 12 h. The as-prepared catalysts were reduced using pure hydrogen at 250 °C for 2 h, and then passivated with 1% O$_2$/Ar at room temperature for another 30 min to prevent bulk oxidation.

**Catalyst characterization.** The specific surface area and total pore volume of the CNT were measured via N$_2$ physisorption using a Micrometrics ASAP 2020 at −196 °C. The amounts of oxygen-containing groups on the CNT support were analyzed by a Pekin-Elmer thermal analyzer at a heating rate of 10 °C min$^{-1}$ to 800 °C under nitrogen atmosphere. Raman spectra were collected on a Horiba Jobin Yvon Labram HR instrument. HAADF-STEM images were obtained using a Tecnai G2 F20 S-Twin transmission electron microscope with an accelerating voltage of 200 kV. CO chemisorption was carried out using an Autochem-II 2920 analyzer (Micromeretics) equipped with a thermal conductivity detector (TCD). The aberration-corrected HAADF-STEM images were obtained using an FEI Titan Themis 300 STEM system operating at 200 kV equipped with spherical aberration (Cs) correctors. EPR measurements were conducted on a Bruker EMX spectrometer. X-ray photoelectron spectroscopy (XPS) measurements were conducted on a Kratos XSAM-800 scanning X-ray microprobe with an Al K$_\alpha$ (hυ = 1486.6 eV) X-ray as the excitation source. The C 1s peak at 284.6 eV was taken as an internal standard to correct the shift in the binding energy caused by sample charging. Electronic conductivity was tested by a four-probe resistivity tester. The XAFS spectra at Pt L$_3$ (E$_0$ = 11,564.0 eV) edge was performed at BL14W1 beamline of Shanghai Synchrotron Radiation Facility (SSRF) operated at 3.5 GeV under "top-up" mode with a constant current of 240 mA. The XAFS data centers under fluorescence mode with a Lytle ion chamber. The energy was calibrated according to the absorption edge of a pure Pt foil. The experimental absorption coefficients as a function of energies μ(E) were processed using background subtraction and normalization procedures, and reported as "normalized absorption" with E$_0$ = 11,564.0 eV for all the tested samples and Pt foil standard. Quasi in situ XPS spectra were recorded on a VG MultiLab 2000 spectrometer with an Omicron Sphera II hemispherical electron energy analyzer with a monochromatic Al K$_\alpha$ X-ray source (1486.6 eV, anode operating at 15 kV and 300 W), in which the constant pass energy and base pressure of the system were set as 40 eV and 5.0 × 10$^{-9}$ mbar, respectively. The catalysts were treated under the real reaction conditions with elevated temperature in a homemade reaction chamber under ambient pressure, and then transferred to the XPS analysis chamber for XPS measurement under 10$^{-9}$ Torr pressure and room temperature through a load-lock gate without exposure to air.

**Catalyst testing.** The CO oxidation of these catalysts was carried out in a continuous flow U-shaped reactor under atmospheric pressure. Prior to the reaction, the as-prepared Pt/CNT-X catalysts were sieved to a particle size of 0.075–0.106 mm. Then, 20 mg of the catalyst was placed on a quartz wool preloaded in the reactor. A feed stream containing 1 vol.% CO as well as 20 vol.% O$_2$ and Ar as balance was fed to the initial catalyst with a space velocity (S.V.) of 60,000 mL·g$_{cat}^{-1}$·h$^{-1}$. The catalyst was directly exposed to the reaction gas without any further pretreatment. The influent and effluent gases were analyzed using an online Agilent 7890 gas chromatograph (GC) equipped with a thermal conductivity detector (TCD). Each data point was taken at an interval of 2 h. Moreover, both the external and internal diffusion limitations in this reactor under these conditions were investigated via Mears and Weisz-Prater analyses (Supplementary Information). For the Pt/CNT-600 catalyst with the highest catalytic activity, the calculated values under kinetics conditions were 0.014 for the Mears criterion for external diffusion, and 0.007 for the Weisz-Prater criterion for internal diffusion, which enabled us to exclude the influences of external and internal diffusion limitations on the activity in this study.

**DFT calculations.** All DFT calculations were performed using the Vienna ab initio simulation package (VASP)[48–51] within the generalized gradient approximation

(GGA) using Perdew–Burke–Ernzerhof (PBE) functional[52]. The DFT-D3 correction scheme[53] was used to evaluate the weak van der Waals interactions between the adsorbed molecules and the catalyst surface[54]. We used a cutoff energy of 400 eV for Kohn–Sham orbitals, a Monkhorst-Pack grid for the k-points sampling in the Brillouin zone and a second-order Methfessel-Paxton smearing with a width of 0.05 eV[55]. We used a 10-atom Pt cluster on graphene to model the above catalysts for a qualitative comparison of the electronic influences by OCGs incorporation. All the supported Pt$_{10}$ cluster models were based on the graphene substrate built using periodic slab geometry with a 5 × 3√3 × 1 supercell (12.30 Å × 12.78 Å × 15.00 Å). As the number of OCGs over carbon support surface was in much excess with respect to that of Pt particles for the above Pt catalysts, it is most likely that the Pt particles nucleate and grow over the OCGs. To investigate the effects of OCGs, the hydroxyl, carboxyl, carbonyl, and ester were separately incorporated into the graphene to construct four OCGs-incorporated substrate-supported Pt10 clusters (Pt-hydroxyl, Pt-carboxyl, Pt-carbonyl, and Pt-ester). A 2 × 2 × 1 Monkhorst-Pack k-point mesh within the surface Brillouin zones was used for these models. In all models, geometry optimizations were conducted by using a force-based conjugated gradient method[56]. The transition states of the elementary steps were located by means of the dimer method. Convergence of saddle points and minima was believed to reach when the maximum force in each degree of freedom was less than 0.03 eV · Å$^{-1}$. To obtain the atomic charges, a fast algorithm operating on a charge density grid was carried out for Bader charge analysis[57,58]. The charge density difference is calculated as ρ(r) = ρ$_{total}$(r) − ρ$_{Pt}$(r) − ρ$_{support}$, where ρ$_{total}$(r), ρ$_{Pt}$(r), and ρ$_{support}$ are the electron densities of the Pt supported on the support, Pt cluster and support, respectively. The resultant charge density difference was plotted by using a VESTA visualization software.

**Transient kinetic analysis.** For the transient kinetic analysis, the experiments were conducted in the same U-shaped reactor. Similarly, 20 mg of the catalyst was preloaded into the reactor and subjected to a reactant mixture of 1 vol.% CO, 20 vol.% O$_2$ and Ar as balance with an S.V. of 60,000 mL·g$_{cat}^{-1}$·h$^{-1}$. The feed flow was replaced by a flow of Ar after the reaction steady state was reached under 100 °C, while maintaining the flow rate of Ar before and after the switch. After reaching stability, the feed gas was switched to Ar + C$^{16}$O + $^{16}$O$_2$ (>99.9%), Ar + C$^{16}$O + $^{16}$O$_2$ (50%)/$^{18}$O$_2$ (50%), and Ar + C$^{16}$O + $^{18}$O$_2$ (>97%) with P$_{CO}$: P$_{O2}$:P$_{Ar}$ = 1:20:79 at an S.V. of 60,000 mL·g$_{cat}^{-1}$·h$^{-1}$. The effluent gas stream was monitored using a Balzers QMG 422 quadrupole mass spectrometer (MS).

**Steady-state isotopic transient kinetic analysis (SSITKA).** For SSITKA, the experiments were conducted in the same U-shaped reactor. Similarly, 20 mg of the catalyst was preloaded into the reactor, which was then filled with quartz wool to minimize the dead volume. The catalysts were first subjected to a reactant mixture of 1 vol.% CO, 20 vol.% O$_2$ and Ar as balance with an S.V. of 60,000 mL·g$_{cat}^{-1}$·h$^{-1}$. The feed flow was replaced by a flow of Kr/$^{13}$CO/O$_2$ or Kr/CO/$^{18}$O$_2$ after reaction steady-state was reached under 100 °C and 1.85 bar, while maintaining the S.V. and the partial pressures of CO and O$_2$ constant. In situ CO adsorption was performed using an isotopic switch from 1 vol.% $^{12}$CO and Ar as balance to 1 vol.% $^{13}$CO and Kr as balance at the same conditions. Once the isotope exchange was complete, the inlet gas mixture was switched back to $^{12}$CO/Ar, and this operation was repeated for several times to calculate the total number of active sites (N$_{total}$). The effluent gas stream was monitored using a Balzers QMG 422 quadrupole mass spectrometer (MS) and an Agilent 7890 gas chromatograph (GC).

The average surface residence time of species i, τ$_{i,measured}$, could be calculated based on the area under the response curve:

$$\tau_{i,measured} = \int_0^\infty F_i(t) dt \tag{3}$$

The average surface residence time, τ$_{i,corrected}$, was further corrected based on the residence of Ar during the switch:

$$\tau_{i,corrected} = \tau_{i,measured} - \tau_{Ar} \tag{4}$$

Thus, the number of adsorbed species (N$_i$) could be calculated from the average surface residence time (τ$_{i,measured}$) and the exit flow (F$_{i, exit}$) of species i:

$$N_i = \tau_{i,corrected} \cdot F_{i,exit} \tag{5}$$

The corresponding surface coverage (θ$_i$) could be calculated as:

$$\theta_i = \frac{N_i}{N_{total}} \tag{6}$$

## Data availability

The authors declare that all the important data to support the findings in this paper are available within the main text or in the Supplementary information. Extra data are available from the corresponding author upon reasonable request

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

## Acknowledgements

W.C., X.Z., and X.D. acknowledge funding from the Natural Science Foundation of China (21922803, 92034301, 22008066, and 21776077), the China Postdoctoral Science Foundation (BX20190116), the Innovation Program of Shanghai Municipal Education Commission, the Program of Shanghai Academic/Technology Research Leader (21XD1421000), 111 Project of the Ministry of Education of China (B08021). The authors thank beamline BL14W1 (Shanghai Synchrotron Radiation Facility) for the beam time and assistant in the experiments.

## Author contributions

W.C., D.C., and X.D. conceived this work. W.C. performed the experiments, collected the data, and wrote the paper. D.C. and X.D. designed the research, supervised experiments, and edited the paper. J.Y. and J.Z. conducted the SSITKA and isotopic labeling analysis. J.C. and Y.C. conducted the density-functional theory calculation and analysis. H.Z. and Z.J. helped with XANES spectrometry analyses. G.Q. assisted in the HAADF-STEM and XPS characterization. X.Z. and W.Y. helped with data analyses and discussions. All the authors contributed to the paper revisions.

## Competing interests

The authors declare no competing interests.
