## [Peer Review File · Nature Communications]

Title: Molecular-level insights into the electronic effects in platinum-catalyzed carbon monoxide oxidationREVIEWER COMMENTS

Reviewer #1 (Remarks to the Author):

The effect of the electronic properties on the adsorption and activation of reactants is an important phenomenon for understanding catalytic reactions mechanistic. Establishing a clear relationship between electronic properties of a catalyst and its activity is a challenging and a long-standing issue in the heterogenous catalysis. In this regard, this study is timely and valuable. The authors conducted a systematic and a solid investigation using a variety of in situ tools and theoretical modeling. They experimentally elucidate the influences of Pt electronic properties on (CO, O₂) reactants adsorption properties, especially in operando conditions. Indeed, the activation of these reactants was correlated with Pt electronic properties (Pt charge). I believe that one of the strong assets of this study is the investigation of this phenomenon in situ under realistic catalytic condition of the CO oxidation reaction. Therefore, this manuscript brings sound results on the effect of electronic properties on CO oxidation reaction over Pt catalysts at the molecular-level. In my opinion it deserves to be published in Nature Communications.

1. The caption of Figure 4 is unclear (XPS). The caption said “ambient pressure” while the pressure in the experiment chamber shows 20 torr. The ambient pressure stands for atmospheric pressure conditions and temperature (~1 bar, RT). In general, the information on the in situ XPS experiments should be more precise! They should state clearly the conditions of XPS experiments (P and T).
2. The binding energy shift of Pt 4f_{7/2} was attributed to “shift is mainly ascribed to the variation in Pt electron density due to the electron transfer between Pt and CNT”. What is the driving force for this effect? It would have been interesting to investigate this phenomenon through the examination of the molecular orbitals involved.
3. Although I understand the challenge of recording XPS spectra in in situ conditions, using a laboratory setup (not a synchrotron-based one where high photon brilliance exists), a main critic concerns the in situ XPS spectra especially Pt 4f (more specifically Fig. S9). They were acquired with low counts. During the experiment, they should have been accumulated with more spectra in order to improve the resolution. Figure S9 is not good! The convolution of the pics is quite arbitrary! I am surprised to see such poorly conducted XPS experiment.
4. The figure 4a is irrelevant and does not bring any new information. The technical details shown are obvious and rather standard. It can be removed or displaced to the Supplementary document.
5. Regarding the minimization of the Pt—CO bonding strength, the author stated that “...from the point view of theoretical calculation, the positively charged Pt weakens the normally strong CO adsorption...”. This statement is probably not obvious if we consider that a strong Pt—CO bonding may result from the electron retro-donation from Pt d states to CO molecular orbitals. A Pt positively charged Pt may on the contrary promote CO adsorption. Since this statement is important in the author’s discussion, they should emphasize this effect.
6. The author stated that “Similarly, the employment of Pt B.E. could also help derive the expression of TOF' in good agreement with the experimental TOF' as shown in Fig. S27”. It is not obvious how the BE

was used to derive the TOF. The authors should provide more details regarding this statement.

Reviewer #2 (Remarks to the Author):

In this nice study, Chen and colleagues vary the charge on Carbon-supported Pt nanoparticles and measure the effect on the CO oxidation activity. Similar demonstrations of the effect of charge transfer on the catalytic activity have been reported over the past decade, but the detailed characterization of the Pt electronic structure is the key strength of this paper.

The paper however has many shortcomings that need to be fixed before publication.

A kinetic model links activity to conditions (T, pressures). This paper does not contain a kinetic study. There is no “breakthrough in microkinetic modeling” or a “nanokinetics model”

The use of the term volcano is misleading in this context. In catalysis, volcano curves link activity to a thermodynamic property (adsorption energy, charge, d-band center,...). Here, activity is plotted as a function of the preparation temperature.

It would be nice if the electronic properties of the various carbon supports could be characterized as well, e.g., via C XANES, C XPS, C NMR, or conductivity measurements.

What is the ratio between the number of Pt particles and the number of O defects? Does every particle nucleate at a defect?

In addition to a CO isotopic switch to determine the number of active sites, H₂ pulse chemisorption and H₂-D₂ exchange should be used. Since the CO adsorption energy is sensitive to the Pt charge, the CO isotopic switch might behave differently on the 6 catalysts.

It is unfortunate that no kinetic evaluation has been performed. Light-off curves are generally poor measures of reaction kinetic and are affected by heat and mass transfer. A few measurements around 100C, at limited conversion, and for a range of CO and O₂ partial pressures would provide much more information than the data reported here. The orders in CO and O₂ would be particularly interesting.

The coverage in the DFT calculations does not match the SSITKA coverages, and the analysis assumes a model that is first order in CO and O₂. This is not correct. The correlation between the CO + O₂ adsorption energy and the Bader charge is misleading, as the CO and O₂ adsorption energy respond very differently to charge (see SSITKA data). CO adsorption generally weakens with charge, O₂ adsorption strengthens. From Figure 2f, it seems the effect is largest on O₂.

The absence of O₂ isotope scrambling (p 11) does not prove non-dissociative adsorption. It proves that reaction of O* with CO* is faster than reaction of O* with O*.

The relation between the CO and the oxygen coverage and the Pt charge measured by SSITKA results from reaction kinetics, not only from the adsorption energy. Moreover, based on theory, one would expect a linear relation between charge and adsorption energy, and hence an exponential relation between coverage and charge.

The manuscript is at times difficult to follow because of poor grammar.

Many thanks for the valuable comments and suggestions from the two reviewers. We have revised our paper by fully taking into account all the comments and suggestions.

Reviewer #1 (Remarks to the Author):

The effect of the electronic properties on the adsorption and activation of reactants is an important phenomenon for understanding catalytic reactions mechanistic. Establishing a clear relationship between electronic properties of a catalyst and its activity is a challenging and a long-standing issue in the heterogenous catalysis. In this regard, this study is timely and valuable. The authors conducted a systematic and a solid investigation using a variety of in situ tools and theoretical modeling. They experimentally elucidate the influences of Pt electronic properties on (CO, O₂) reactants adsorption properties, especially in operando conditions. Indeed, the activation of these reactants was correlated with Pt electronic properties (Pt charge). I believe that one of the strong assets of this study is the investigation of this phenomenon in situ under realistic catalytic condition of the CO oxidation reaction. Therefore, this manuscript brings sound results on the effect of electronic properties on CO oxidation reaction over Pt catalysts at the molecular-level. In my opinion it deserves to be published in Nature Communications.

1. The caption of Figure 4 is unclear (XPS). The caption said “ambient pressure” while the pressure in the experiment chamber shows 20 torr. The ambient pressure stands for atmospheric pressure conditions and temperature (~1 bar, RT). In general, the information on the in situ XPS experiments should be more precise! They should state clearly the conditions of XPS experiments (P and T).

Response:

We are very sorry for this mistake about the pressure of XPS measurement. Indeed, the quasi in situ XPS equipment is made up of two chamber: the reaction chamber is working at ambient pressure, while the analysis chamber working under 10⁻⁹ torr pressure. To make it more clearly, we have revised the relevant description as following:

“The catalysts were treated under the real reaction conditions with elevated temperature at a homemade reaction chamber under ambient pressure, and then transferred to the XPS analysis chamber for XPS measurement under 10⁻⁹ torr pressure and room temperature through a load-lock gate without exposure to air.”

2. The binding energy shift of Pt 4f_{7/2} was attributed to “shift is mainly ascribed to the variation in Pt electron density due to the electron transfer between Pt and CNT”. What is the driving force for this effect? It would have been interesting to investigate this phenomenon through the examination of the molecular orbitales involved.

Response:

Thanks for this good question. The electronic effects of the oxygen-containing groups (OCGs) of CNT on the supported metal particles could be categorized into two types: i) the inductive effects, involving the polarization of bonds owing to the differences in the electronegativity; ii) the resonance effects, involving the actual movement of electrons through a π -bond system (*Organic Pharmaceutical Chemistry. In Remington: The Science and Practice of Pharmacy, 21st ed.; Troy, D. B., Ed.; Lippincott Williams & Wilkins: Philadelphia, PA, 2005; Chapter 25, pp 386–409*). Typically, an electron-donating group (EDG) is a functional group that donates some of its electron density to a conjugated π system via the resonance (+R) or inductive effect (+I), whereas an electron-withdrawing group (EWG) removes electron density via the resonance (-R) or inductive effect (-I). Hence, the driving force for this electronic effect is mainly arising from the inductive/resonance effects (*Chem. Mater., 2015, 27, 7362-7369*).

To make it more clearly, we have conducted atomic orbital analysis based on the suggestions of the reviewer, and added the relevant description in the revised version as following:

“Considering the high electron conductivity of CNT (Table S4) in terms of the high percentage of sp^2 -hybridized carbon with respect to sp^3 -hybridized carbon (Fig. S10) to neutralize the initial ion charge, the observed binding energy shift is mainly attributed to the electron transfer between Pt and CNT. Because the Pt 4f level is not too deep and the wave-function overlap between the Pt 4f and 5d levels is significant, the variation in the Pt 5d states occupation number yields more pronounced effects in the determination of binding energy shift than that of total number of electrons³⁵. As a result, both XANES and XPS analyses confirm the significant loss of 5d state electrons of Pt over CNT-600 upon bonding/hybridization with OCGs from the inductive/resonance effects³⁷, which is vice versa for Pt over CNT-0.”

3. Although I understand the challenge of recording XPS spectra in in situ conditions, using a laboratory setup (not a synchrotron-based one where high photon brilliance exists), a main critic concerns the in situ XPS spectra especially Pt 4f (more specifically Fig. S9). They were acquired with low counts. During the experiment, they should have been accumulated with more spectra in order to improve the resolution. Figure S9 is not good! The convolution of the pics is quite arbitrary! I am surprised to see such poorly conducted XPS experiment.

Response:

Thanks very much for kindly reminding us on this issue. According to your valuable suggestion, we have accumulated with more spectra to improve the resolution of Figure S9, and the results are still in good consistence with the current analyses on the Pt electronic properties. Hence, we have replaced Figure S9 with the new one in the revised version as following:

Figure S9. Typical XPS Pt 4f spectra of (a) Pt/CNT-0, (b) Pt/CNT-200, (c) Pt/CNT-400, (d) Pt/CNT-600, (e) Pt/CNT-800 and (f) Pt/CNT-1000.

4. The figure 4a is irrelevant and does not bring any new information. The technical details shown are obvious and rather standard. It can be removed or displaced to the Supplementary document.

Response:

We fully agree with the reviewer's good suggestion. Hence, we have removed Figure 4a in the revised version, and the new Figure 4 has been shown as following:

Figure 4. In situ XPS Pt 4f spectra of Pt/CNT-600 under the reaction atmosphere at elevated temperature.

5. Regarding the minimization of the Pt—CO bonding strength, the author stated that “...from the point view of theoretical calculation, the positively charged Pt weakens the normally strong CO adsorption...”. This statement is probably not obvious if we consider that a strong Pt—CO bonding may result from the electron retro-donation from Pt d states to CO molecular orbitals. A Pt positively charged Pt may on the contrary promote CO adsorption. Since this statement is important in the author's discussion, they should emphasize this effect.

Response:

We are sorry for not emphasizing this effect of electron retro-donation on the adsorption of CO. As the reviewer suggested, a strong Pt-CO bonding may result from the electron retro-donation from Pt d states to CO molecular orbitals. Typically, a decrease of the electron density of Pt results in a decrease of the back-donation of the metal electrons into $2\pi^*$ antibonding orbitals of the CO molecule and a strengthening of the C-O bond, but not necessarily a weakening of the Pt-CO bond (*J. Phys. Chem. B*, 2005, 109, 23430-23443; *Surf. Sci.* 1998, 396, 156-175). Therefore, there is no clear relationship between the charge of Pt and the adsorption of CO. Based on the above analysis, we have revised the relevant discussion as following:

“Notably, the above result is still speculative because of the lack of direct evidence on the activation and adsorption of reaction species under operando conditions. Moreover, it has been suggested in previous study that the changes of the electronic properties of metal may result in an increase of electron back-donation, but not necessarily a strengthening of the M-CO bond¹². Therefore, in situ kinetics information to bridge the microscopic electronic properties of the Pt active site with the macroscopic catalytic performance of CO oxidation were obtained as follows.”

6. The author stated that “Similarly, the employment of Pt B.E. could also help derive the expression of TOF' in good agreement with the experimental TOF' as shown in Fig. S27”. It is not obvious how the BE was used to derive the TOF. The authors should provide more details regarding this statement.

Response:

Thanks for the reviewer's kind suggestion. In the revised version, we have added the details on the derivation of TOF based on Pt B.E. in the Supporting Information as following:

4. The derivation of TOF based on Pt B.E.

As shown in Fig. S31a, $\ln A_i$ exhibits an almost linear dependence on E_a , indicating a significant kinetic compensation effect. According to Cremer-Constable relation, this relation could be interpreted as a thermodynamic balance between the activation entropy and activation enthalpy. Based on the transition state theory, these two kinetics parameters serve as the representations of the adsorption and activation of reaction species, respectively. For

this reason, the activation properties of reactants (E_a) could be well correlated with Pt electronic properties (Pt B.E.) as shown in Fig. S31b. On the other hand, to experimentally elucidate the influences of Pt electronic properties on reactants adsorption properties, steady-state isotopic transient kinetic analysis (SSITKA) was conducted to determine the site coverage of surface species and the intrinsic reaction rate under operando conditions. Typically, the reaction temperature and pressure were raised to 100 °C and 1.85 bar, respectively, which were maintained for another 2 h prior to measuring the reaction rate (r_{100}') and TOF' in Fig. S31c and Table S7. Specifically, the ^{12}CO response curves by switching from $\text{Ar}/^{12}\text{CO}/\text{O}_2$ to $\text{Kr}/^{13}\text{CO}/\text{O}_2$ were recorded in Fig. S27, based on which the amount of adsorbed CO (N_{CO}) as well as site coverage (θ_{CO}) could be calculated in Table S7. Moreover, the O_2 site coverages (θ_{O_2}) were investigated by switching from $\text{Ar}/\text{CO}/^{16}\text{O}_2$ to $\text{Kr}/\text{CO}/^{18}\text{O}_2$ as shown in Fig. S28, which exhibits the almost identical $^{16}\text{O}_2$ and Ar response curves possibly due to the limited O_2 adsorption on the catalyst surface. However, based on the formation of OOCO from adsorbed CO and O_2 to produce CO_2 (Scheme 1) and the direct reaction between the adsorbed O and CO to produce CO_2 , the amount of adsorbed oxygen species, mainly O_2 , could be estimated from the amount of generated CO_2 due to the large excess of CO over Pt surface. As a result, the oxygen site coverages (θ_{oxygen}) based on C^{16}O_2 response curves by switching from $\text{Ar}/\text{CO}/^{16}\text{O}_2$ to $\text{Kr}/\text{CO}/^{18}\text{O}_2$ (Fig. S29) were calculated as shown in Table S7. As a result, the adsorption behaviors of reaction species (θ_{CO} and θ_{oxygen}) were further correlated with Pt B.E. in Fig. S31d. Obviously, θ_{CO} decreases with Pt B.E., agreeing well with the decreased adsorption energy by Pt Bader charge, while θ_{oxygen} exhibits an opposite trend. Hence, these in situ kinetics information give a quantitative description of CO and oxygen adsorption with Pt B.E., consistent with the positively charged Pt weakening the adsorption of CO to provide more active sites for oxygen adsorption as indicated by DFT calculations.

According to the above DFT calculations, the rate-determining step for CO oxidation for these Pt catalysts appears to be formation of OOCO specie from adsorbed CO and O_2 . Based on this, the catalytic activity of active site (TOF') could be estimated by the expression of $\text{TOF}' = k' \times \theta_{\text{CO}} \times \theta_{\text{oxygen}}$, in which k' is the reaction rate constant. Further combining the Arrhenius equation, $k' = A_0' \times \exp(-E_a/RT)$, yields Eq. S1:

$$\ln A_0' = \ln TOF' - \ln \theta_{CO} - \ln \theta_{oxygen} + E_a / RT \quad (S1)$$

As a result, $\ln A_0'$ (logarithm of frequency factor) could be calculated as shown in Table S7, and further correlated with Pt B.E. in Fig. S31e. According to the transition-state theory, $\ln A_0'$ is proportional to the activation entropy (ΔS^{\ddagger}), which mainly quantifies the freedom loss of reactants species in terms of binding strength with catalyst. To this point, because Pt B.E. could give a good description of the reactants activation behaviors (Fig. S31b) and adsorption behaviors (Figs. S31d and S31e), it is attempted to combine the influences of Pt B.E. on the activation and adsorption to derive a new kinetics model to bridge the microscopic properties of Pt active site and the macroscopic catalytic performance. Herein, combining the above linear relationships gives Eq. S2:

$$TOF' = a \times \exp(b \times \text{Pt B.E.} + c) \quad (S2)$$

in which a , b , and c is determined in Table S8 at the given condition. As a result, the predicted TOF' (solid line) based on Eq. 2 exhibits good consistence with the experimental TOF' (red triangle) in Fig. 31f.

Figure S31. The relationship between (a) $\ln A_i$ and E_a , as well as (b) E_a , (c) r_{100}' , (d) θ_{CO} and θ_{oxygen} , (e) $\ln k_0'$, (f) the experimental and predicted TOF' as a function of Pt B.E. for Pt/CNT-0, Pt/CNT-200, Pt/CNT-400, Pt/CNT-600, Pt/CNT-800 and Pt/CNT-1000.

Reviewer #2 (Remarks to the Author):

In this nice study, Chen and colleagues vary the charge on Carbon-supported Pt nanoparticles and measure the effect on the CO oxidation activity. Similar demonstrations of the effect of charge transfer on the catalytic activity have been reported over the past decade, but the detailed characterization of the Pt electronic structure is the key strength of this paper.

The paper however has many shortcomings that need to be fixed before publication.

1. A kinetic model links activity to conditions (T, pressures). This paper does not contain a kinetic study. There is no “breakthrough in microkinetic modeling” or a “nanokinetics model”

Response:

We are very sorry to use these exaggerated words, and we have revised them as following:

“Here, we report an unconventional kinetics strategy of bridging the microscopic metal electronic structure and the macroscopic steady-state rate for CO oxidation over Pt catalysts.”

“In this regard, based on (in situ) spectroscopy/microscopy, isotopic labelling, DFT calculations and in situ kinetics analysis, we present an unconventional kinetics strategy of bridging this “materials gap” in traditionally kinetic analysis to correlate the microscopic properties of active site with the macroscopic catalytic performance as schematically depicted in Fig. 5”

“Further incorporating catalyst structural parameters yields a new model for quantifying the electronic effects and predicting catalytic performance.”

“This in situ kinetic strategy to identify reaction pathway and kinetic indicator to establish the new model could predict the catalytic performances and be extended to the design of other metal catalysts.”

“Obviously, this model links the electronic and structural properties of Pt active sites in nanoscale with the macroscopic catalytic performance”

2. The use of the term volcano is misleading in this context. In catalysis, volcano curves link activity to a thermodynamic property (adsorption energy, charge, d-band center,...). Here, activity is plotted as a function of the preparation temperature.

Response:

We are sorry to use this misleading word, and we have revised it as following:

“then a combination of isotopic labelling technique and DFT calculations is employed to identify the reaction pathway as well as kinetic indicator by further correlating with the catalytic performance”

“Taking Pt^0 $4f_{7/2}$ spin-orbit peak for example, the corresponding Pt B.E continuously increases from 71.60 eV (Pt/CNT-0) to 71.90 eV (Pt/CNT-600), followed by a decline to 71.74 eV (Pt/CNT-1000) as depicted in Fig. 1b.”

3. It would be nice if the electronic properties of the various carbon supports could be characterized as well, e.g., via C XANES, C XPS, C NMR, or conductivity measurements.

Response:

Thanks very much for the reviewer’s kind suggestion. We have conducted C 1s XPS and electronic conductivity measurements, and added the relevant description in the revised version as following:

“Considering the high electron conductivity of CNT (Table S4) in terms of the high percentage of sp^2 -hybridized carbon with respect to sp^3 -hybridized carbon (Fig. S10) to neutralize the initial ion charge, the observed binding energy shift is mainly attributed to the electron transfer between Pt and CNT.”

Figure S10. Typical XPS C 1s spectra of Pt/CNT-0, Pt/CNT-200, Pt/CNT-400, Pt/CNT-600, Pt/CNT-800, and Pt/CNT-1000.

Table S4. Electronic conductivity of Pt/CNT-0, Pt/CNT-200, Pt/CNT-400, Pt/CNT-600, Pt/CNT-800 and Pt/CNT-1000.

Catalyst	Electronic conductivity (S/cm)
Pt/CNT-0	5.3
Pt/CNT-200	9.7
Pt/CNT-400	12.8
Pt/CNT-600	13.4
Pt/CNT-800	14.2
Pt/CNT-1000	15.2

4. What is the ratio between the number of Pt particles and the number of O defects? Does every particle nucleate at a defect?

Response:

Thanks for the reviewer's good question. Based on the XPS Pt 4f and O 1s spectra, the molar ratio of O to Pt, $n_{\text{O}}/n_{\text{Pt}}$, is determined to be 61.5, 42.3, 39.0, 31.8, 24.6 and 19.3 for Pt/CNT-0, Pt/CNT-200, Pt/CNT-400, Pt/CNT-600, Pt/CNT-800 and Pt/CNT-1000, respectively. For the ~ 1nm Pt particle, it has around 35 Pt atoms (*J. Am. Chem. Soc.* 2014, 136, 16736-16739). In this regard, the ratio of the number of O defects and the number of Pt particles could be estimated as 1076, 740, 682, 556, 430 and 338. It can be seen that the number of O defects is in much excess with respect to that of Pt particles. Assuming the homogenous distribution of O defects and Pt particles around CNT external surface, it is very likely that every Pt particle nucleates at more than one O defect. In the revised version, we have added the relevant description as following:

"Because the number of OCGs over carbon support surface was in much excess with respect to that of Pt particles for the above Pt catalysts, it is most likely that the Pt particles nucleate and grow over the OCGs. To investigate the effects of OCGs, the hydroxyl, carboxyl, carbonyl, and ester were separately incorporated into the graphene to construct four OCGs-incorporated

substrate-supported Pt₁₀ clusters (Pt-hydroxyl, Pt-carboxyl, Pt-carbonyl, and Pt-ester).”

5. In addition to a CO isotopic switch to determine the number of active sites, H₂ pulse chemisorption and H₂-D₂ exchange should be used. Since the CO adsorption energy is sensitive to the Pt charge, the CO isotopic switch might behave differently on the 6 catalysts.

Response:

According to the reviewer’s suggestion, we have conducted H₂ pulse chemisorption measurement, and the results are shown in Table S2. It can be seen that the amount of H₂ adsorption is higher than that of Pt atoms for each catalyst, which could be attributed to H spillover to carbon support (J. Catal. 1979, 58, 287–295), H diffusion into the bulk (Surf. Sci. 1985, 160, 37–45), or the ability of under-coordinated metal atoms present at the edges and corners of supported particles to bind more than one H (J. Catal. 1972, 24, 367–384). These results are also consistent with the high dispersion of Pt particles over carbon support, and we have added the relevant description in the revised version as following:

“The average Pt particle size (d_{Pt}) for these catalysts was determined to be 1.2–1.3 nm by averaging 200 random particles (Fig. 1b), and the highly dispersed Pt particles were confirmed by the H₂-chemisorption results in Table S2.”

Table S2. The structural and electronic properties, as well as catalytic activity of Pt/CNT-0, Pt/CNT-200, Pt/CNT-400, Pt/CNT-600, Pt/CNT-800 and Pt/CNT-1000 for CO oxidation (100 °C, atmospheric pressure, P_{CO}:P_{O₂}:P_{Ar}=1:20:79).

Catalyst	Pt B.E. (eV)	r_{100} (mmol _{CO} ·mol _{Pt} ⁻¹ ·s ⁻¹)	d_{Pt} (nm) ^a	N_{tot,H_2} (μmol·g _{cat} ⁻¹) [*]	TOF _{Pt} *10 ⁻³ (s ⁻¹) ^b	$N_{tot,CO}$ (μmol·g _{cat} ⁻¹)	d_{CO} (nm) ^c	TOF _{CO} *10 ⁻³ (s ⁻¹) ^d
Pt/CNT-0	71.60	8.6	1.2	52.11	13.4	17.76	1.3	14.7
Pt/CNT-200	71.68	16.6	1.3	61.21	28.8	19.77	1.2	25.4
Pt/CNT-400	71.85	32.8	1.2	63.91	51.1	23.17	1.1	42.7
Pt/CNT-600	71.90	35.6	1.2	60.68	55.5	21.88	1.1	49.1
Pt/CNT-800	71.80	26.6	1.3	58.31	46.2	17.63	1.3	45.5
Pt/CNT-1000	71.74	24.2	1.2	59.73	37.7	23.36	1.1	31.2

^a determined from HAADF-STEM measurement.

^b based on the Pt particle size from HAADF-STEM measurement.

^c determined from the ¹²CO-¹³CO isotopic switches at 100 °C.

^d based on the reversible adsorption of CO from the ¹²CO-¹³CO isotopic switches at 100 °C.

*Note: It can be seen that the amount of H₂ adsorption is higher than that of Pt atoms for each catalyst, which could be attributed to H spillover to carbon support (J. Catal. 1979, 58, 287–295), H diffusion into the bulk (Surf. Sci. 1985, 160, 37–45), or the ability of under-coordinated metal atoms present at the edges and corners of supported particles to bind more than one H (J. Catal. 1972, 24, 367–384). These results are also consistent with the high dispersion of Pt particles over carbon support.

6. It is unfortunate that no kinetic evaluation has been performed. Light-off curves are generally poor measures of reaction kinetic and are affected by heat and mass transfer. A few measurements around 100 °C, at limited conversion, and for a range of CO and O₂ partial pressures would provide much more information than the data reported here. The orders in CO and O₂ would be particularly interesting.

Response:

Thanks for this kind suggestion. We have measured the dependence of reaction rate on the partial pressures of CO and O₂, and added the relevant description in the revised version:

“Moreover, the reaction orders of CO and O₂ for the most positively charged Pt/CNT-600 and negatively charged Pt/CNT-0 were measured as shown in Fig. S30. It is obvious that Pt/CNT-0 exhibits much lower CO reaction order of -0.58 compared with that of -0.10 for Pt/CNT-600, consistent with its higher CO site coverage. On the other hand, the almost same O₂ reaction orders around 0.9 further evidence the much lower O₂ site coverages for these catalysts.”

Figure S30. The kinetic reaction orders of (a) CO and (b) O₂ for Pt/CNT-0 and Pt/CNT-600.

7. The coverage in the DFT calculations does not match the SSITKA coverages, and the analysis assumes a model that is first order in CO and O₂. This is not correct. The correlation between the CO + O₂ adsorption energy and the Bader charge is misleading, as the CO and O₂ adsorption energy respond very differently to charge (see SSITKA data). CO adsorption generally weakens with charge, O₂ adsorption strengthens. From Figure 2f, it seems the effect is largest on O₂.

Response:

Thanks for the reviewer's kindly reminding us on the coverage in the DFT calculation mismatching the SSITKA coverage. We have carried out more DFT calculations to study the effects of CO coverage on the reaction energy, in which the CO-preadsorbed models were constructed in Fig. S24. As a result, the energy barrier for the formation of OOCO complex from adsorbed CO and O₂ could be calculated as shown in Fig. S25a. It is found that the

energy barrier still demonstrate an almost linear decline with Pt Bader charge in Fig. S25b. Accordingly, we have added the relevant description in the revised version as following:

“Moreover, the effects of CO coverage on the reaction energy were investigated, in which the corresponding CO-preadsorbed configurations were constructed in Fig. S24. Accordingly, the energy barrier for the formation of OOCO complex from adsorbed CO and O₂ could be calculated as shown in Fig. S25a, which still demonstrates an almost linear decline with Pt Bader charge in Fig. S25b.”

Figure S24. Optimized CO-preadsorbed configurations of all reaction intermediates involved in the pathway of OOCO formation on (a) Pt-basal, (b) Pt-hydroxyl, (c) Pt-carboxyl, (d) Pt-carbonyl, and (e) Pt-ester. (gray: carbon, red: oxygen; blue: platinum; white: hydrogen)

Figure S25. (a) The energy barrier for OOCO formation over CO-preadsorbed Pt-basal, Pt-hydroxyl, Pt-carboxyl, Pt-carbonyl, and Pt-ester. (b) The relationships between the energy barrier (ΔE_i) and Pt Bader charge.

Moreover, based on the above analysis, it is found that the rate-determining step remains to be the formation of OOCO complex from adsorbed O_2 and CO. Because the reaction order for each reactant in an elementary step is equal to its stoichiometric coefficient, the reaction orders for CO and O_2 in this rate-determining step are determined to be 1. To make it more clearly, we have revised the description as following:

“According to the above DFT calculations, the rate-determining step for CO oxidation for these Pt catalysts involves the formation of OOCO species from adsorbed CO and O_2 (Fig. 2d). Because the reaction order for each reactant in an elementary step is equal to its stoichiometric coefficient, the reaction orders of CO and O_2 in this rate-determining step ($CO^* + O_2^* \rightarrow OOCO^* + *$) are determined to be 1.”

Lastly, the reviewer is right that the correlation between the CO + O₂ adsorption energy and the Bader charge is misleading, because the CO and O₂ adsorption energy respond very differently to charge as reflected by SSITKA. Hence, we have removed the correlation in Fig. 2f, and changed the relevant description in the revised version as following:

“Considering that the adsorption of reactants, specifically CO, has been suggested as the key factor for this reaction, the effects of Pt charge on their adsorption were further compared in Fig. S26. Evidently, Pt-hydroxyl demonstrates the highest adsorption energy for the reactants. This is consistent with a previous study showing that the strong adsorption of CO severely poisons Pt active sites, and a general consensus to improve the activity relies on weakening the Pt-CO bonding for CO desorption to provide more active sites for O₂ activation⁴³.”

8. The absence of O₂ isotope scrambling (p 11) does not prove non-dissociative adsorption. It proves that reaction of O* with CO* is faster than reaction of O* with O*.

Response:

We are sorry for this inaccurate description. Based on the detection of ¹⁶O¹⁸O isotopologues from equimolecular ¹⁶O₂/¹⁸O₂ mixtures, the absence of that suggests the sluggish dissociation/association of oxygen, because the quasi-equilibrated O₂ dissociation forms binomial isotopologue distributions (50% ¹⁶O¹⁸O) (*J. Catal.* 2012, 285, 92–102). In the revised version, we have revised the description as following:

“Because the quasi-equilibrated O₂ dissociation forms binomial isotopologue distributions (50% ¹⁶O¹⁸O)⁴¹, the absence of that suggests the sluggish dissociation/association of oxygen. ”

9. The relation between the CO and the oxygen coverage and the Pt charge measured by SSITKA results from reaction kinetics, not only from the adsorption energy. Moreover, based on theory, one would expect a linear relation between charge and adsorption energy, and hence an exponential relation between coverage and charge.

Response:

Thank you very much for the reviewer’s kindly reminding us on this issue. As the reviewer stated, one would expect an exponential relation between coverage and charge if there exists a

linear relation between charge and adsorption energy. Hence, we have correlated the site coverage of CO and oxygen with Pt charge and Pt B.E. by exponential functions in the revised version, and the results could be still employed to derive the expression of TOF as shown in Figs. 3g and 31f. Hence, we have revised the description as following:

“Similarly, inspired by the linear dependence of adsorption energy (E_{ads}) on Pt Bader charge (Fig. 2f), the adsorption behaviors of the reaction species (θ_{CO} and θ_{oxygen}) were further correlated with the Pt charge by exponential functions (Fig. 3e).”

Figure 3. Isotopic labelling studies and SSITKA of CO oxidation. *a*, Mass spectrometry (MS) data collected for the Pt/CNT-600 catalyst during the switch from Ar to Ar+C¹⁶O+¹⁶O₂ (>99.9%), Ar+C¹⁶O+¹⁶O₂ (50%)/¹⁸O₂ (50%), and Ar+C¹⁶O+¹⁸O₂ (>97%) at an ambient pressure. *b*, The relationship between logarithm of frequency factor ($\ln A_i$) and activation energy (E_a). E_a (*c*), r_{100} (*d*), site coverages of CO (θ_{CO}) and oxygen (θ_{oxygen}) (*e*), logarithm of frequency factor ($\ln A_o'$) (*f*),

as well as the experimental and predicted TOF' (g) as a function of Pt charge for Pt/CNT-0, Pt/CNT-200, Pt/CNT-400, Pt/CNT-600, Pt/CNT-800, and Pt/CNT-1000. Reaction conditions: 100 °C, $P_{CO}:P_{O_2}:P_{Ar}=1:20:79$, $60000 \text{ mL}\cdot\text{g}_{\text{cat}}^{-1}\cdot\text{h}^{-1}$, and 1.85 bar.

10. The manuscript is at times difficult to follow because of poor grammar.

Response:

In the revised version, we have obtained helps from Editage (www.editage.com) for English language editing, and improved the language as possible as we can. We hope it meet the standard of the publication.

REVIEWERS' COMMENTS

Reviewer #1 (Remarks to the Author):

The authors have taken my comments into account and made the recommended changes. The manuscript has been considerably improved as far as I am concerned. I think that this manuscript can be published.

Reviewer #2 (Remarks to the Author):

The authors have addressed my questions and somewhat improved the analysis, but some inconsistencies remain. Since the data are interesting, I recommend publication.

Additional suggestions:

Question 3: It is counterintuitive that the conductivity of the support (e.g., for CNT-1000 vs CNT-600) does not correlate with the XPS shifts and the derived Pt charges. One would expect that a higher conductivity results from a higher free carrier concentration in the support and hence results in an increased charge transfer to the Pt particles, as is the case with typical semiconductor supports. This is not the case here and the connection between support electronic properties and the catalyst activity is somewhat indirect.

Question 6. The reduction in the CO order is consistent with weaker CO adsorption on Pt/CNT-600. The reduction in the O₂ order for Pt/CNT-600 is consistent with stronger O₂ adsorption on Pt/CNT-600. The relation between charge and adsorption energy is hence opposite for both reactants, in line with the SSITKA results (Figure 3e). The DFT analysis misses this point, since only CO and O₂ co-adsorption is considered (Figure 2d), and the kinetic analysis compares the energy of the TS with the energy of co-adsorbed CO and O₂ (energy barrier in Figure 2f). The kinetically relevant energy difference is however between the TS, and adsorbed CO (CO*, zero to negative order) and gas phase O₂ (nearly first order). It is unfortunate that the authors do not use this kinetic information to improve the DFT calculations. Pt/CNT-600 is more active because (largely) the reaction is less hindered by strong CO adsorption. The weaker CO adsorption also explains the lower measured activation energy.

Reviewer #1 (Remarks to the Author):

The authors have taken my comments into account and made the recommended changes. The manuscript has been considerably improved as far as I am concerned. I think that this manuscript can be published.

Response:

We appreciate the reviewer for the positive comments and the previous valuable suggestions.

Reviewer #2 (Remarks to the Author):

The authors have addressed my questions and somewhat improved the analysis, but some inconsistencies remain. Since the data are interesting, I recommend publication.

Additional suggestions:

Question 3: It is counterintuitive that the conductivity of the support (e.g., for CNT-1000 vs CNT-600) does not correlate with the XPS shifts and the derived Pt charges. One would expect that a higher conductivity results from a higher free carrier concentration in the support and hence results in an increased charge transfer to the Pt particles, as is the case with typical semiconductor supports. This is not the case here and the connection between support electronic properties and the catalyst activity is somewhat indirect.

Response:

Thanks for this valuable question. Indeed, the electric conductivity of carbon materials has been widely observed to increase with the annealing temperature ascribed to the elimination of oxygen-containing groups (*Carbon*, 2013, 59, 2-32; *Carbon*, 2016, 96, 174-183; *Carbon*, 2019, 147, 27-34), which is consistent with the trend in Table S4.

Moreover, for semiconductor supports, the impurities (doping) changes the number of charge carriers (electrons or holes) and therefore changes the Fermi level, resulting in increased mobile negative charge carriers in the conduction band or positive charge carriers in the valence band (*Cowper, P., 2017. Azulene-eee en el ell.*). Similarly, in this work, the thermal treatment of oxidized support changed the relative concentrations of electron-withdrawing groups (EWG) and electron-donating groups (EDG), which would decrease and increase the electron density of Pt, respectively. In this regard, we also made correlations between the Pt B.E., Pt charge and

the molar ratio of EWG to EDG (n_{EWG}/n_{EDG}) as depicted in Fig. 1f, and further the catalytic activity in Fig. 2c. To make it more clearly, we have revised the relevant description as following:

Table S4. Electronic conductivity of Pt/CNT-0, Pt/CNT-200, Pt/CNT-400, Pt/CNT-600, Pt/CNT-800 and Pt/CNT-1000.

Catalyst	Electronic conductivity	
	(S/cm)	
Pt/CNT-0	5.3	
Pt/CNT-200	9.7	
Pt/CNT-400	12.8	
Pt/CNT-600	13.4	
Pt/CNT-800	14.2	
Pt/CNT-1000	15.1	

*Note: It can be seen that the electric conductivity of carbon support increases with temperature of heat treatment ascribed to the elimination of oxygen-containing groups (Carbon 2013, 59, 2–32; Carbon 2016, 96, 174–183; Carbon 2019, 147, 27–34).

Fig. 1f. the relationship between n_{EWG}/n_{EDG} and Pt B.E., Pt charge as well as EPR intensity for Pt/CNT-0, Pt/CNT-200, Pt/CNT-400, Pt/CNT-600, Pt/CNT-800, and Pt/CNT-1000.

Question 6. The reduction in the CO order is consistent with weaker CO adsorption on Pt/CNT-600. The reduction in the O₂ order for Pt/CNT-600 is consistent with stronger O₂ adsorption on Pt/CNT-600. The relation between charge and adsorption energy is hence opposite for both

reactants, in line with the SSITKA results (Figure 3e). The DFT analysis misses this point, since only CO and O₂ co-adsorption is considered (Figure 2d), and the kinetic analysis compares the energy of the TS with the energy of co-adsorbed CO and O₂ (energy barrier in Figure 2f). The kinetically relevant energy difference is however between the TS, and adsorbed CO (CO*, zero to negative order) and gas phase O₂ (nearly first order). It is unfortunate that the authors do not use this kinetic information to improve the DFT calculations. Pt/CNT-600 is more active because (largely) the reaction is less hindered by strong CO adsorption. The weaker CO adsorption also explains the lower measured activation energy.

Response:

Thanks for this constructive comment. We fully agree with the referee that the reduction in the CO order is consistent with the weak adsorption of CO, and the increase in the O₂ order is also consistent with the strong adsorption of O₂. In this regard, we have conducted DFT calculations to gain a fundamental understanding of the effects of Pt charge on the adsorption and activation of reactants. Generally, the gas molecule adsorption ability determines the reaction pathways on the catalyst. Due to the much larger adsorption energy of CO compared with O₂, the Pt surface could be dominantly covered by CO. In this case, the Pt reactive site would be blocked to hinder the ER reaction, in which the coadsorption of CO and O₂ on Pt and the LH reaction should be favored, which has been verified by previous DFT calculations (*Phys. Chem. Chem. Phys.* 2012, 14, 16566–16572). Hence, the LH reaction pathway was chosen to study the energy change across the reaction coordinate for these models, and the results were also in line with the SSITKA results as the referee said.

Moreover, according to the referee's suggestion, it is still interesting to compare the energy change between the LH and ER reaction to incorporate the electronic effects of oxygen-containing groups on reaction pathway. Considering the large computational cost to obtain reliable results, it will be studied in our future work.